# Longitudinal Association between Sarcopenia and Cognitive Impairment among Older Adults in Rural Malaysia

**DOI:** 10.3390/ijerph19084723

**Published:** 2022-04-14

**Authors:** K. Ramoo, Noran N. Hairi, A. Yahya, W. Y. Choo, F. Mohd Hairi, D. Peramalah, S. Kandiben, A. Bulgiba, Z. Mohd Ali, I. Abdul Razak, N. Ismail, N. S. Ahmad

**Affiliations:** 1Centre for Epidemiology and Evidence-Based Practice, Department of Social and Preventive Medicine, Faculty of Medicine, University of Malaya, Kuala Lumpur 50603, Malaysia; karthikramoo84@gmail.com (K.R.); abqariyah.yahya@ummc.edu.my (A.Y.); ccwy@um.edu.my (W.Y.C.); farizah@um.edu.my (F.M.H.); devi@ummc.edu.my (D.P.); priyas@um.edu.my (S.K.); awang@ummc.edu.my (A.B.); 2Faculty of Public Health, Universitas Airlangga, Surabaya City 60115, Indonesia; 3Negeri Sembilan State Health Department (JKNNS), Seremban 70300, Malaysia; drzainudin@gmail.com (Z.M.A.); i_inayah@moh.gov.my (I.A.R.); 4Disease Control Division, Ministry of Health Malaysia, Putrajaya 62590, Malaysia; norliana.ismail@moh.gov.my; 5Mental Health, Injury Prevention, Violence and Substance Abuse Sector, Disease Control Division, Ministry of Health, Putrajaya 62590, Malaysia; nursakinahahmad86@gmail.com

**Keywords:** cognitive impairment, older adults, sarcopenia

## Abstract

Sarcopenia is a condition that is highly prevalent among older adults. This condition is linked to numerous adverse health outcomes, including cognitive impairment that impairs healthy ageing. While sarcopenia and cognitive impairment may share a common pathway, limited longitudinal studies exist to show the relationship between these two conditions. Therefore, this study aimed to examine the longitudinal association between sarcopenia and cognitive impairment. This is a cohort study among older adults residing in Kuala Pilah District, Negeri Sembilan, Malaysia. There were 2404 respondents at the baseline and 1946 respondents at one-year follow-up. Cognitive impairment was determined using Mini-mental State Examination scores. Sarcopenia was identified using the Asian Working Group for Sarcopenia 2019 criteria, gait speed was measured using a 4-meter gait test, handgrip strength was assessed using Jamar handheld dynamometer, and appendicular skeletal muscle mass was measured using bioelectrical impedance analysis. Generalized estimating equation (GEE) was used to determine the longitudinal association between sarcopenia and cognitive impairment, presented as relative risk (RR) and its 95% confidence interval. The prevalence of sarcopenia was 5.0% (95% CI 4.00–5.90), and severe sarcopenia was 3.60% (95% CI 2.84–4.31). Upon adjusting for covariates, older adults with sarcopenia have an 80 per cent increased risk of cognitive impairment compared to those without (RR 1.80; 95% CI 1.18–2.75). Similarly, severe sarcopenia was found to significantly increase the risk of cognitive impairment by 101 per cent in the adjusted model (RR 2.01; 95% CI 1.24–3.27). Our study showed that sarcopenia, severe sarcopenia, low physical activity, depressive symptoms, hearing impairment and chronic pain were associated with a higher risk of cognitive impairment among community-dwelling older adults. Therefore, early intervention to prevent sarcopenia, depressive symptoms, hearing impairment, chronic pain, and higher physical activity among older adults is recommended.

## 1. Introduction

The world’s older population is projected to rise from 524 million in 2010 to 1.5 billion by 2050, with the majority of the rise expected in developing countries [1]. As the aged population increases, the proportion of older adults suffering from disability increases [2], representing a demand in the social and healthcare needs. Two prevalent causes of disability and dependence among older adults are sarcopenia and cognitive impairment [3,4,5,6,7]. Sarcopenia is an age-related muscle loss, with loss of muscle strength and or reduced physical performance [8]. The ageing process plays an essential role in the development and progression of sarcopenia, as ageing is known to cause various alterations in the human body [5,8]. Sarcopenia prevalence varies by definition, age groups and settings [9]. Sarcopenia prevalence using European Working Group on Sarcopenia in Older People (EWGSOP) ranges from 1 to 30 per cent, while another review using Asian Working Group for Sarcopenia (AWGS) 2014 found the prevalence range from 4.1 to 11.5 per cent [10,11]. Sarcopenia is a matter of global concern as it affects more than 50 million people and is expected to rise to 200 million over the next 40 years [12]. In addition, sarcopenia has been related to many adverse health outcomes such as falls, reduced endurance, frailty, decreased mobility, disability and ultimately, death [5].

Cognitive impairment among older adults, disables them from performing the activity of daily living and instrumental activity of daily living at the milder stage and leading to dependency at a severe stage [3,4]. Age is the greatest risk for cognitive impairment, and the prevalence is estimated to rise rapidly as age crosses 65 years [3]. Approximately 47 million people are enduring advanced cognitive impairment worldwide [13]. This number is expected to rise to approximately 75 million in 2030 and 132 million in 2050 [13]. 

Many studies found that sarcopenia and cognitive impairment are associated [14,15]. This may suggest that both sarcopenia and cognitive impairment may share a common pathway [16,17,18]. However, there is a knowledge gap in such association as the biological process undermining the relationship between sarcopenia, and cognitive impairment is still unclear [14]. Many studies examining this association are cross-sectional, while longitudinal studies are limited. Thus, this study aimed to examine the longitudinal association between sarcopenia and cognitive impairment.

## 2. Materials and Methods

### 2.1. Study Setting and Design

This study is a part of a cohort study conducted in community-dwelling older adults in Kuala Pilah District, Negeri Sembilan, in 2013. Kuala Pilah district is one of seven districts in Negeri Sembilan, with 14.5 per cent of the population comprising older adults [19]. Two waves of data collection with 12 months elapsed between wereundertaken involving the older adults age 60 and above. The first wave began in 2013 and was completed in 2014, while the second wave was carried out from 2014 to 2015. The study protocol has been previously published [20]. A total of 2404 participants completed the questionnaires at the first wave, and a total of 1946 were followed up during the second wave.

### 2.2. Sampling Strategy

Participant’s recruitment was accomplished using a two-stage sampling method utilizing the sampling frame from the 2010 National Population Census Report by the Department of Statistic, Malaysia (DoS). For the first stage, DoS has randomly selected 156 enumeration blocks (EB) out of a total of the 254 EB’s in the Kuala Pilah District. As for the second stage, 16 living quarters (LQs) out of the whole of 80 to 120 living quarters were randomly selected by DoS using a computer-generated list.

### 2.3. Participant’s Eligibility Criteria

The inclusion criteria are individuals age 60 and above, residing in Kuala Pilah district for at least 12 months before the study and Malaysian nationals. The age of the participants was verified using their date of birth stated on the identity card. Excluded participants were non-Malaysians, those not at home at the time of the interview, and those who were unable to communicate independently. The interviews and sarcopenia components measurements were carried out in the participant’s homes by four teams of four trained personnel each team, headed by a physician. Each personnel received geriatric assessment training by the physician before data collection.

### 2.4. Data Collection

#### 2.4.1. Sarcopenia Assessment

Sarcopenia was defined based on the AWSG 2019 algorithm. Participants with low appendicular skeletal muscle mass index with low handgrip strength or low gait speed were defined as having sarcopenia, while participants with low appendicular skeletal muscle mass index, low handgrip strength and low gait speed were defined as having severe sarcopenia [8].

#### 2.4.2. Muscle Mass Measurement

A 4-point bioelectrical impedance analysis (BIA) equipment (TANITA TBF-300A, Tanita, Japan) at 50kHz used to measure impedance. Participants were asked to stand barefoot on the electrodes with the fore foot placed directly on the front sole electrode and the heel placed directly on the rear sole electrodes. Appendicular skeletal muscle mass (ASM) was calculated using gender-specific equations; ASM (for men) = 0.197 × (impedance index) + 0.179 × (weight) − 0.019 (R2 = 0.87, standard error of the estimate = 0.98 kg) and ASM (for women) = 0.221 × (impedance index) + 0.117 × (weight) + 0.881 (R2 = 0.89, standard error of the estimate = 0.81 kg) [21]. The AWGS 2019 criteria of 7.0kg/m^2^ in men and 5.7kg/m^2^ in women were used as cut-offs for ASM/ht2 [8].

#### 2.4.3. Grip Strength, Gait Speed and Anthropometry Measurement

Grip strength assessment was accomplished utilizing a dynamometer (JAMAR hand dynamometer 5030JI; Sammons Preston, Bolingbrook, IL, USA). Cut-offs for grip strength were based on the AWGS 2019 criteria: <28 kg in men and <18 kg in women. Four-meter gait speed assessment was performed on the flat surface at the participant’s home. The cut-off of <1.0 m/s was used to measure low gait speed (8). Height was measured to the nearest 0.1 cm, with a stadiometer (SECA 217, Hamburg, Germany). Weight was measured to the nearest 0.1 kg using the bioelectrical impedance analysis (BIA) equipment described above.

#### 2.4.4. Cognitive Impairment Measurement

Cognitive impairment status was determined using the Mini-Mental State Examination (MMSE). The Malay version of the MMSE has been locally validated by Zarina et al. 2007, [22]. The education level adjusted MMSE cut-off scores is <25 for uneducated participants, <27 for individuals with primary level education and <29 for secondary or above level education [23].

#### 2.4.5. Other Variables

Depressive symptoms and physical activity were measured using the locally validated Geriatric Depression Scale (GDS) and Physical Activity Scale for Elderly (PASE) [24,25]. Other measures include age, gender, chronic pain, hearing and visual impairment. Age was categorized into three groups: age 60–69, 70–79 and age 80 and above. The responses of hearing and visual impairment are categorized into yes and no. Chronic pain was categorized into yes and no, recorded using a validated question among older Malaysian that has been used in various geriatric studies [20,26].

### 2.5. Statistical Analysis

Data from this study were analyzed using SPSS software version 23 (IBM SPSS Statistic 23, Armonk, NY, USA) with a *p*-value < 0.05 to be considered statistically significant. Participant’s sociodemographic details presented as a categorical variable reported in frequency and prevalence while chi-square test was used to test for differences in categorical variables. Multiple imputations (MI) were applied to the variables with more than 5% of missing data to attenuate the bias arising from the missing data. The variable with more than 5 per cent missing data are low gait speed, low handgrip strength, and low ASM index. The longitudinal association between sarcopenia and cognitive impairment was determined using generalized estimating equation (GEE) analysis. The respondent with the outcome at the baseline was excluded from the analysis. Relative risk was derived using the modified Poisson regression approach in the GEE analysis [27]. The Poisson loglinear model was selected for the distribution and link function. The unstructured working correlation structure was selected as it provided the lowest Quasi-likelihood information criterion (QIC) value. In addition, unstructured correlation structure allows for all possible correlations. The association between exposure and outcome were measured by conducting both the unadjusted and adjusted GEE models. In the adjusted analysis, the risk factors identified as apriori were forced into the model. Goodness of fit of the final model was determined by computing the Corrected Quasi-likelihood under Independence Model Criterion (QICC). Post study sampling weight was applied to correct for the over-representation or under-representation of the target population. Design effect was taken into consideration upon the sample size calculation.

### 2.6. Ethical Consideration

The Medical Research & Ethics Committee (MREC) (NMRR-20-3065-57439) approved the study. Written informed consent was obtained from all participants prior to data collection.

## 3. Results

The sociodemographic characteristic of this study participant is shown in Table 1. Most of the study participants were at the age of 60–79 (86.0%). The majority of the study participants were women, accounting for 62.4 per cent of the participants. Nearly one third of the participants presents with depressive symptoms and visual impairment while less than one fourth of the participants presents with chronic pain and hearing impairment.

Table 2 shows the prevalence of sarcopenia and severe sarcopenia. From this study, the prevalence of sarcopenia and severe sarcopenia was 5.0 and 3.6 per cent, respectively. Upon stratification by gender, men were observed to have a higher prevalence of sarcopenia and severe sarcopenia than women. The majority of the participants were found to have low gait speed (96.0 per cent), while more than half were found to have low handgrip strength (60.2 per cent). However, a very small prevalence of participants was observed to have low appendicular skeletal muscle mass (ASM) index (5.1 per cent). Low ASM index prevalence was higher in men than women, while the prevalence of low handgrip strength and low gait speed were higher among women than men.

The effect of sarcopenia and severe sarcopenia on cognitive impairment were shown in Table 3. In the unadjusted model, sarcopenia significantly increase the risk of cognitive impairment by 113 per cent. Upon adjusting for covariates, the effect remains significant where older adults with sarcopenia have 80 per cent increased risk of cognitive impairment compared to those without. Similarly, severe sarcopenia was found to significantly increase the risk of cognitive impairment by 185 per cent in the unadjusted model. Upon adjusting for the covariates, older adults with severe sarcopenia have a 101 per cent increased risk of developing cognitive impairment in one year follow up compared to those without. In addition, the covariates comprising low physical activities represented by the lowest PASE quartiles, depressive symptoms, chronic pain and hearing impairment were observed to be significantly associated with cognitive impairment.

Further analysis was conducted to examine the effect of sarcopenia components on cognitive impairment, as shown in Table 4. The unadjusted analysis of low gait speed, low handgrip strength and low ASM index on cognitive impairment shows the relative risk of 1.04 (95% CI 0.64–1.69), 2.53 (95% CI 1.94–3.31) and 2.11 (95% CI 1.46–3.04), respectively. However, in the adjusted model, only low handgrip strength and low ASM index were significantly associated with cognitive impairment. Older adults with low handgrip strength and low ASM index have 110 per cent and 81 per cent increased risk of developing cognitive impairment, respectively, in a one-year duration compared to those without. Comparing between models, model with low handgrip strength has the highest risk and the lowest QICC value (not shown in table) among the sarcopenia component. Collinearity diagnostic and interaction assessment were performed, and none was identified.

## 4. Discussion

We have examined the prevalence of sarcopenia and the longitudinal association between sarcopenia and cognitive impairment among community-dwelling older adults. 

The prevalence of sarcopenia from this study was low compared to recent studies using AWSG 2019 criteria within the same region among the rural older population [28,29]. In addition, a recent systematic review demonstrated that sarcopenia prevalence varied between 10 per cent to 27 per cent using 151 studies across the globe [30]. The same review has also demonstrated that severe sarcopenia prevalence ranged between 2 per cent to 9 per cent, indicating severe sarcopenia prevalence from our study was within range. The low prevalence of sarcopenia in this study is due to the low prevalence of the low ASM index. Low ASM index is the primary requirement for the sarcopenia diagnosis as per AWGS 2019 algorithm (8). The low ASM index prevalence from this study was relatively lower compared to other studies [31,32]. Skeletal muscle mass among older adults is affected by several factors, including age, gender, nutrition, physical activities and co-existing comorbidities [33,34,35,36,37]. One plausible reason that resulted in a low prevalence of low ASM index was the effect of physical activities as Kuala Pilah district is a rural district that is extensively involved in agricultural activities from which the participants were sampled [38]. Studies have shown that older adults who are physically active during their midlife have a higher skeletal muscle mass during their old age than those who are not physically active [37].

Our finding shows that sarcopenia and severe sarcopenia is longitudinally associated with cognitive impairment. Specifically, severe sarcopenia exhibits a greater risk of cognitive impairment among older adult in the one-year duration than sarcopenia. 

Findings from previous studies investigating the relationship between sarcopenia and severe sarcopenia with cognitive impairment or cognitive function are in a cross-sectional and longitudinal design, showing mixed results. A recent systematic review and meta-analysis examining the association of sarcopenia and cognition using nine cross-sectional and one longitudinal study from Asian and western countries reported a significant association between sarcopenia and cognitive impairment (pooled OR = 2.50, 95% CI = 1.26–4.92) [14]. In a prospective study among 131 community-dwelling older adults in Japan, sarcopenia was significantly associated with cognitive deterioration measured using MMSE over the one-year duration (OR = 7.86; 95% CI = 1.53–40.5) [39]. Another longitudinal study among 496 older adults in Mexico reported that sarcopenia is significantly associated with mild cognitive impairment (OR = 1.74, 95% CI = 1.02, 2.96) [40]. A British study using a cohort of older British men from the British Regional Heart Study (BRHS), investigated the association of sarcopenia and severe sarcopenia with mild and severe cognitive impairment [41]. However, the reported relative risk from that study was not significant [41]. Likewise, other cross-sectional studies among older adults reported a non-significant association between sarcopenia and cognitive impairment [42,43]. In contrast, other cross-sectional studies reported a significant association between sarcopenia and cognitive impairment [44,45]. The heterogeneity in the findings could be explained by the differences in the tools used to define and measure sarcopenia and cognitive impairment or cognitive function [40].

As for the association between sarcopenia components with cognitive impairment, our study shows low handgrip strength and low ASM index are longitudinally associated with cognitive impairment. Our findings are similar to several longitudinal studies that have investigated low handgrip strength and low skeletal muscle mass with cognitive impairment and cognitive function [41,46,47,48,49]. However, our study shows low gait speed was not longitudinally associated with cognitive impairment. This finding is inconsistent with other longitudinal studies that demonstrated a significant association between low gait speed and cognitive impairment. A Japanese longitudinal study among 1096 community-dwelling older adults found that slow gait was associated with a significant reduction in cognitive function [47]. However, the study had a longer follow-up duration (ten years) compared to our study Finding from two longitudinal studies revealed that slow gait speed predicted the transitioning from mild to severe cognitive impairment ((InCHIANTI: HR: 2.08, 95% CI 1.40–3.07; LASA: HR: 1.33, 95%CI 1.01–1.75) [50]. One possible reason for the inconsistent finding was the duration of follow up. In our study, the follow-up duration was one year, while other studies have duration ranges from 4.4 to 25 years, that has demonstrated a significant association between slow gait speed and cognitive function [47,50]. In addition, post hoc analysis of low gait speed association with cognitive impairment shows there was inadequate power to detect a significant association. Apart from that, other modifiable factors were also found to be significantly increased the risk of cognitive impairment in the fully adjusted model. This includes low physical activities indicated by the lowest PASE quartile, depressive symptoms, chronic pain, and hearing impairment. Hence, early intervention targeting low physical activity, chronic pain, and hearing impairment serves as a risk reduction measure for cognitive impairment along with sarcopenia.

The specific mechanism underlying the association between sarcopenia and cognitive impairment could be explained via the ‘Common Cause Hypothesis’. The ‘Common Cause Hypothesis’ suggests a common factor that drives the non-cognitive and cognitive processes [16,17,18]. Sarcopenia, with its decline in physical function, is linked to cognition via the ‘Common Cause Hypothesis’ [18]. This hypothesis arises primarily from the strong association between decline in physical and cognitive function as reported in the cross-sectional and longitudinal studies [14,18,39]. The common cause of ageing has been implicated in several neurodegenerative changes that reduce physical and cognitive functions [51,52,53,54,55]. Ageing results in sarcopenia via several mechanisms, including neural activation and muscle synergy formation impairment, cortical hypoexcitability, structural connectivity integrity and basal ganglia dysregulation, causing muscle weakness and reduction in gait speed [55]. Ageing leads to a reduction in cognitive function via age-related changes comprising of structural brain changes, age-related synaptic loss, and age-related depletion of nigrostriatal dopamine [51,52,53,54].

The common cause ‘ageing’ process via age-related changes, results in sarcopenia and reduced cognitive functions. With respect to this, evidence shows that sarcopenia precedes cognitive impairment. Findings from two previous longitudinal studies among older adults shows that sarcopenia predicts cognitive impairment or reduction of cognitive function [39,40]. Likewise, previous longitudinal study has also demonstrated that the sarcopenia components, mainly low gait speed and low handgrip strength predicted cognitive impairment or reduction of cognitive function [46,47,48,56,57]. This evidence strongly suggests that sarcopenia and its components could predict cognitive impairment. Based on the above evidence, it is suggestive that age-related loss of muscle mass, function, and strength occurs prior to the reduction of cognition function as evidence shows sarcopenia and its components could predict future cognitive impairment. In addition, recent evidence also suggests that imbalance in myokine secretion and vascular dysfunction links sarcopenia and cognitive impairment [58]. 

Despite most evidence showing sarcopenia increases the risk of cognitive impairment, there is some evidence shows that cognitive impairment increases the risk of sarcopenia. As described by Morley et al. (2021), a decline in brain function could result in a reduction of muscle function such as loss of muscle mass and weakness following a cerebrovascular accident or ‘dual task’ deficit in individuals with dementia that could result in exercise reduction leading to sarcopenia [59]. However, in this study individuals with cognitive impairment at the baseline were excluded, yielding incident cognitive impairment in relation to sarcopenia. In this study, sarcopenia was shown to increase the risk of cognitive impairment. 

Hence early detection and intervention of sarcopenia is essential to reduce the risk of cognitive impairment among older adults. In addition, AWGS 2019 recommends the measurement of possible sarcopenia in the primary health care setting. The AWGS algorithm suggests the measurement of muscle strength or physical performance for the identification of possible sarcopenia. Our study recommends handgrip strength assessment for the identification of possible sarcopenia. This is because it has the highest effect size and the best model fit predicting cognitive impairment. In addition, it also serves as a risk-stratifying method for helping healthcare providers determine poorer cognitive functioning [48]. Handgrip strength measurement can be done in a seated position, reducing the risk of falls among older adults and not requiring ample space in the primary health care setting.

There are several strengths of this study. This is the first study of its kind in Malaysia and as well as in the Southeast Asia region. Second, post hoc analysis of the association between sarcopenia and severe sarcopenia with cognitive impairment shows that the samples were adequately powered. Third, this study uses a longitudinal analysis taking into account the time-varying confounders using imputed data sets to attenuate possible bias arising from the missing data. The important confounders adjusted include depressive symptoms and physical activity as addressed by Chang et al. 2016 [60]. Apart from that, measurement error was reduced by using education corrected MMSE score as MMSE is biased by education level. Variables such as age, gender, visual, and hearing impairment were adjusted in the analysis, which helps to improve the precision of the estimate [61]. There are several limitations to this study. First, impedance measurement using BIA is affected by water content. However, participants were advised to consume adequate water before performing the test. Second, the duration of exposure variable was unknown, especially sarcopenia, depressive symptoms, and chronic pain, and therefore, the effect of exposure duration on cognitive impairment cannot be examined. Third, the study population is rural population and therefore caution needs to apply to generalize this finding to general population. Lastly, this study has a short follow up duration and therefore the long-term effect of sarcopenia on cognitive impairment cannot be examined.

## 5. Conclusions

In conclusion, sarcopenia increases the risk of cognitive impairment among community-dwelling older adults. Early identification and prompt intervention of sarcopenia, depressive symptoms, chronic pain, hearing impairment and low physical activity serves as risk reduction measures for cognitive impairment.

## Figures and Tables

**Table 1 ijerph-19-04723-t001:** Baseline sociodemographic details of the study participants.

Variables	Total *n*%	Cognitive Impairment Present, *n* (%)	Cognitive Impairment Absent, *n* (%)	Chi-Square Value (*p*-Value)
Age				115.1 (<0.001)
60–69	1103 (46.9)	442 (37.9)	661 (55.8)	
70–79	918 (39.1)	483 (41.5)	435 (36.7)	
≥80	329 (14)	240 (20.6)	89 (7.5)	
Gender				101.7 (<0.001)
Male	884 (37.6)	320 (27.4)	564 (47.6)	
Female	1467 (62.4)	846 (72.6)	621 (52.4)	
PASE score				118.4 (<0.001)
First Quartile	431 (19.7)	310 (28.9)	121 (10.8)	
Second Quartile	633 (28.9)	291 (27.1)	342 (30.6)	
Third Quartile	551 (25.2)	246 (22.9)	305 (27.3)	
Fourth Quartile	574 (26.2)	226 (21.1)	348 (31.3)	
Depressive symptoms				3.7 (0.01)
Yes (≥6)	763 (32.5)	397 (34.1)	366 (30.9)	
No (≤5)	1583 (67.5)	766 (65.9)	817 (69.1)	
Chronic pain				21.7 (<0.001)
Yes	524 (22.3)	307 (26.3)	217 (18.3)	
No	1826 (77.7)	859 (73.7)	967 (81.7)	
Visual Impairment				43.2 (<0.001)
Yes	935 (39.8)	542 (46.5)	393 (33.2)	
No	1412 (60.2)	623 (53.5)	789 (66.8)	
Hearing Impairment				60.5 (<0.001)
Yes	366 (15.6)	250 (21.5)	116 (9.8)	
No	1981 (84.4)	915 (78.5)	1066 (90.2)	

**Table 2 ijerph-19-04723-t002:** Prevalence of Sarcopenia and its component status at baseline, overall and stratified by gender.

	*N*	Overall % (95% CI)	Men % (95% CI)	Women % (95% CI)
Sarcopenia	120	5.0 (4.0–5.9)	8.5 (6.6–0.3)	2.8 (1.9–3.6)
Severe Sarcopenia	87	3.6 (2.8–4.3)	5.7 (4.2–7.3)	2.3 (1.5–3.1)
Low ASM Index ^1^	123	5.1 (4.0–5.9)	8.8 (6.8–10.7)	2.8 (2.0–3.5)
Low HG Strength ^2^	1447	60.2 (58.2–62.1)	54.6 (51.0–57.5)	63.6 (61.2–66.1)
Low Gait Speed	2308	96.0 (95.2–96.8)	92.3 (90.8–97.0)	98.2 (97.6–98.9)

^1^ ASM: appendicular skeletal mass, ^2^ HG: handgrip.

**Table 3 ijerph-19-04723-t003:** GEE analysis showing longitudinal analysis of sarcopenia and severe sarcopenia with cognitive impairment.

Exposure	b	Unadjusted RR (95% CI)	*p* Value	b	Adjusted RR ^1^ (95% CI)	*p* Value
**Sarcopenia Status**						
Sarcopenia	0.76	2.13 (1.46–3.09)	<0.001	0.59	1.80 (1.18–2.75)	0.01
Severe Sarcopenia	1.05	2.85 (1.95–4.15)	<0.001	0.77	2.01 (1.24–3.27)	<0.001

^1^ Adjusted model, adjusted for age, gender, physical activity, depressive symptoms, chronic pain, visual and hearing impairment. Full table presented in Appendix A.

**Table 4 ijerph-19-04723-t004:** GEE analysis showing longitudinal effect of sarcopenia components with cognitive impairment.

Exposure	b	Unadjusted RR (95% CI)	*p* Value	b	Adjusted RR ^1^ (95% CI)	*p* Value
Low Gait Speed						
No		1			1	
Yes	0.04	1.04 (0.64–1.69)	0.88	−0.21	0.81 (0.48–1.38)	0.44
Low Handgrip Strength						
No		1			1	
Yes	0.93	2.53 (1.94–3.31)	<0.001	0.74	2.1 (1.58–2.79)	<0.001
Low ASM Index ^2^						
No		1			1	
Yes	0.74	2.11 (1.46–3.04)	<0.001	0.60	1.81 (1.2–2.73)	0.01

^1^ Adjusted model, adjusted for age, gender, physical activity, depressive symptoms, chronic pain, visual and hearing impairment. ^2^ ASM: Appendicular skeletal mass.

## Data Availability

The datasets generated and analyzed during the current study can be obtained from the first author or corresponding author on reasonable request.

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
