# Peer review of "Longitudinal Association between Sarcopenia and Cognitive Impairment among Older Adults in Rural Malaysia"

_ijerph, 2022, doi:10.3390/ijerph19084723_

Round 1

Reviewer 1 Report

I consider that in principle the work meets the standards that imply the corresponding elaboration of a quality scientific approach. The topic is interesting and relevant. The aspects of the Introduction and the Discussions are correctly argued and strengthened by indicating recent and quality bibliographic sources. I suggest that the authors add a brief description of the link and significance of the results between the methods used in points 2.4.2 to 2.4.5.

Reviewer 2 Report

Thank you for this presentation which I read with interest. 

The manuscript aimed to examine the longitudinal association between sarcopenia and cognitive impairment. The presented study combines data from several disciplines, which has an added value.

However, I have a few minor doubts about the manuscript in its current form. These will be discussed in detail below with the aim of proposing improvements.

Author Response

Point 1: The brand names of the tools used are listed, but in one case it is not there (name of the body composition measurement equipment), please complete it.

Response 1:

Dear Reviewer

Thank you for the comments

The name of the body composition measurement tool was -TANITA TBF-300A, Tanita, Japan. I have added to subsection 2.4.2 in the first sentence.

Point 2: Were there any exclusion criteria for the study? Eg contraindications to bioelectrical impedance analysis (BIA)?

Response 2: No other exclusion criteria other than explained in subsection 2.3. As for the contraindication of BIA, the commonly stated conditions for BIA contraindication/restriction are pregnancy and cardiac implants. Since most of the study population consisted of older adults aged 60 and above and therefore pregnancy criteria were not considered. As for cardiac implants, a study by Chabin et al, (2019) showed that BIA is safe for individuals with cardiac implants as no significant changes in battery voltage, lead impedance, or pacing thresholds were found during the BIA assessment.

Chabin, X., Taghli-Lamallem, O., Mulliez, A., Bordachar, P., Jean, F., Futier, E., . . . Eschalier, R. (2019). Bioimpedance analysis is safe in patients with implanted cardiac electronic devices. Clinical Nutrition, 38(2), 806-811. doi:10.1016/j.clnu.2018.02.029.

Point 3: What were the reasons for not joining the second part of the study? Of course, probably the death of the participants, but only? Please complete the text with a short sentence.

Response 3: The second part is related to the mortality of the participants and regarding the second part of the study I have arranged for a longer follow-up duration (later censored date) to assess the all-cause mortality risk associated with sarcopenia, severe sarcopenia, low gait speed, low handgrip strength and low ASM index.

Thank you

Reviewer 3 Report

Thank you for an interesting paper. I have the following comments:

  • Regarding sarcopenia assessment, the authors stated that”Participants with low appendicular skeletal muscle mass index with low handgrip strength or low gait speed were defined as having sarcopenia, while participants with low appendicular skeletal muscle mass index, low handgrip strength and low gait speed were defined as having severe sarcopenia”.

The cut-off values between sarcopenia and severe sarcopenia in terms of low appendicular skeletal muscle mass index, low handgrip strength or low gait speed are needed to be identified.

-           In conclusion, the authors stated that sarcopenia increases the risk of cognitive impairment among community-dwelling older adults. However, the design of this study does not enable a true causal association to be ascertained. Though sarcopenia increases the risk of cognitive impairment among community-dwelling older adults, it is also possible that cognitive impairment may lead to sarcopenia.

The authors need to debate more on this limitation in the discussion section.

Author Response

Point 1: The cut-off values between sarcopenia and severe sarcopenia in terms of low appendicular skeletal muscle mass index, low handgrip strength or low gait speed are needed to be identified.

Response 1:

Dear Reviewer,

Thank you for the comments, The cut-off values in terms of low appendicular skeletal muscle mass index, low handgrip strength, and low gait speed were described in method subsections 2.4.2 and 2.4.3.

Point 2: In conclusion, the authors stated that sarcopenia increases the risk of cognitive impairment among community-dwelling older adults. However, the design of this study does not enable a true causal association to be ascertained. Though sarcopenia increases the risk of cognitive impairment among community-dwelling older adults, it is also possible that cognitive impairment may lead to sarcopenia. The authors need to debate more on this limitation in the discussion section.

Response 2:

I have added into the discussion section regarding the possible pathway of cognitive impairment leading to sarcopenia. Marked using track changes in the manuscript.

Thank you

This manuscript is a resubmission of an earlier submission. The following is a list of the peer review reports and author responses from that submission.